# Efficacy and Safety of Clonidine in the Treatment of Acute Mania in Bipolar Disorder: A Systematic Review

**DOI:** 10.3390/brainsci13040547

**Published:** 2023-03-25

**Authors:** Prakamya Singal, Nicolas A. Nuñez, Boney Joseph, Leslie C. Hassett, Ashok Seshadri, Balwinder Singh

**Affiliations:** 1Department of Psychiatry, All India Institute of Medical Sciences, New Delhi 110029, India; 2Department of Psychiatry & Psychology, Mayo Clinic, Rochester, MN 55905, USA; 3Department of Neurology, Mayo Clinic, Rochester, MN 55905, USA; 4Mayo Clinic Libraries, Mayo Clinic, Rochester, MN 55905, USA

**Keywords:** clonidine, systematic review, acute mania, bipolar disorder

## Abstract

Clonidine, an alpha-2 adrenergic agonist, has been proposed as an antimanic agent that acts by reducing noradrenergic transmission. We conducted a systematic review to examine the efficacy and safety of clonidine for acute mania/hypomania. A comprehensive literature search was performed to identify randomized controlled trials (RCT) and non-randomized studies investigating the efficacy and safety of monotherapy/adjuvant treatment with clonidine for acute mania/hypomania in patients with bipolar disorder (BD). Nine studies (*n* = 222) met our inclusion criteria, including five RCTs (*n* = 159) and four non-randomized studies (*n* = 63). Non-randomized studies showed clonidine to help reduce symptoms of mania. However, data from placebo controlled RCTs were inconsistent. One RCT showed adjuvant clonidine as superior to placebo, whereas another RCT reported that clonidine was not better than placebo. In individual RCTs, lithium and valproate offered better antimanic effects compared to clonidine. Studies reported hypotension, depression, and somnolence as common adverse effects. Significant differences in study design and sample size contributed to high heterogeneity. This systematic review suggests low-grade evidence for clonidine as an adjuvant treatment for acute mania with mood stabilizers and inconclusive efficacy as monotherapy, warranting further well-designed RCTs.

## 1. Introduction

Bipolar disorder (BD) affects 1%–2% of the general population, with a lifetime prevalence rate of 0.4%–2.4% [1,2]. BD is characterized by episodes of mania or hypomania, alternating with severe depression [3,4]. These episodes are frequent and persistent and significantly impact the socio-occupational functioning of patients with BD [5]. The pharmacological management of acute manic episodes involves using atypical antipsychotics such as olanzapine, risperidone, aripiprazole, quetiapine, ziprasidone, and cariprazine or mood stabilizers such as lithium, sodium valproate, and carbamazepine [6,7,8,9]. Although atypical antipsychotics and mood stabilizers have good efficacy in treating acute episodes, both classes of drugs are associated with significant side effects such as metabolic syndrome, tardive dyskinesia, and extrapyramidal symptoms with atypical antipsychotics [10].

An imbalance in noradrenergic activity at the neuronal synaptic cleft has been proposed as one of the hypotheses for the causation of mania [11]. The noradrenergic hyperactivity theory suggests increased activity in the central sympathetic system, abnormal cortical and thalamic dopaminergic/noradrenergic and serotonin production, and altered sensitivity of alpha-2 and beta-2 adrenergic receptors [12,13]. Clonidine, an alpha-2 adrenergic agonist, has been explored as an anti-manic agent in patients with BD [14,15,16]. Clonidine primarily targets presynaptic alpha-2 receptors in the locus coeruleus, reducing the release and turnover of norepinephrine, thus, potentially relieving symptoms of mania [17]. Despite the relevant biological mechanism, clonidine’s efficacy for acute mania remains equivocal [18]. Clonidine is used to treat a wide spectrum of clinical disorders such as anxiety, hypertension, opioid withdrawal, neonatal abstinence syndrome, attention deficit hyperactivity disorder, tic disorder, chronic pain associated with cancer, vasomotor symptoms associated with menopause, and hyperhidrosis [17,19]. Clonidine is a well-established centrally acting anti-hypertensive agent that mechanistically works by inhibiting the synaptic release of norepinephrine [20] with modest effects on serotonin and dopamine receptors. With the advancement in the understanding of neurobiology of psychiatric disorders, there is an interest in the field to repurpose older drugs that are safe and have good tolerability [21]. A recent randomized controlled trial (RCT) showed promising results with adjuvant clonidine use for mania among inpatients with BD [22], thus, reinvigorating interest in clonidine’s use as a treatment option for acute mania.

With this review, we aim to systematically appraise the evidence on the efficacy, safety, and tolerability of clonidine monotherapy/adjunctive treatment for acute mania.

## 2. Methods

A protocol was developed for this systematic review following the guidance of the Preferred Reporting Items for Systematic Reviews and Meta-Analyses [23]. A protocol has been registered and approved in open science framework (Registration number: https://doi.org/10.17605/OSF.IO/3EUXM).

### 2.1. Data Sources and Search Strategies

A comprehensive search of several databases was performed on 20 January 2023. No language or date limits for the search were applied. Databases searched (and their content coverage dates) were Ovid MEDLINE^®^ (1946+ including epub ahead of print, in-process, and other nonindexed citations), Ovid Embase (1974+), Ovid Cochrane Central Register of Controlled Trials (1991+), Ovid Cochrane Database of Systematic Reviews (2005+), Web of Science Core Collection via Clarivate Analytics (1975+), and Scopus via Elsevier (1970+). The search strategies were designed and conducted by a medical librarian (LCH) with input from the study investigators. Controlled vocabulary supplemented with keywords was used to search for studies that examined the efficacy and safety of clonidine for management of the acute mania. References of potentially eligible articles were also reviewed to expand the search. The actual strategy listing all search terms used and how they are combined is available in the Appendix A.

### 2.2. Study Selection

Two reviewers (PS and BS) working independently and in pairs identified and screened the titles and abstracts of studies that met the inclusion criteria. Our inclusion criteria were (1) Population: adult (≥18 years) patients in an acute mania/hypomania or mixed state; (2) Intervention: pharmacological treatment with clonidine (monotherapy or adjunct); (3) *Control*: placebo or control or treatment as usual; and (4) *Outcome*: reported outcome data on remission and response rates, all-cause of discontinuation, and change in the severity of symptoms of mania assessed by the use of standardized behavioral scales (such as Brief Psychiatric Rating Scale [BPRS] or Young Mania Rating Scale [YMRS]) from baseline to endpoint and reported side effects. Studies were selected irrespective of the language. We included RCTs, non-randomized studies, and observational studies. Case series, case reports, and reviews were excluded from the systematic review.

Relevant articles were included in the Covidence software and used for screening of titles, abstracts, and full text review. Any disagreement between the reviewers was resolved by consensus. Included articles were qualitatively analyzed to determine efficacy, tolerability, dose range, duration of treatment, and study bias.

### 2.3. Data Collection

Data from the included studies were extracted by two reviewers (PS and NAN) using a standardized data extraction form. We extracted data on study characteristics (author, year, country, study design, sample size, demographic characteristics of study patients, inclusion and exclusion criteria, and conclusion), intervention types, outcome measures, and follow-up. The primary outcome measure was a change in acute mania/hypomania/mixed symptom severity measured using standardized rating scales between the beginning and end of the treatment intervention period as well as reported outcome data on remission and response rates.

### 2.4. Methodological Quality and Risk of Bias Assessment

The risk of bias was assessed for the RCTs using the Cochrane Collaboration’s risk of bias tool [24] assessing following domains: sequence generation, allocation concealment, blinding of study participants and personnel, blinding of the outcome assessment, selective reporting, and incomplete outcome data and other biases. Risk of bias was designated to be high if described protocols were concerning for bias in a domain. If description of the domain was omitted from the primary text, the risk of bias was labeled as “unknown.” When an adequate protocol was described for a given domain, it was labeled “low risk.” For the non-randomized studies, the methodological quality was assessed using Methodological Index for Non-Randomized Studies [25,26,27].

## 3. Results

### 3.1. Study Selection

A total of 587 abstracts were screened, of which 20 articles were selected for full-text review. Nine studies reporting change in manic symptom severity post-clonidine met the inclusion criteria for the systematic review. Five RCTs comparing clonidine to placebo/control (lithium or verapamil) [22,28,29,30,31], and four non-randomized studies—three open-label studies [14,15,32] and one double blind study [33]—were included (Figure 1).

### 3.2. Characteristics of the Included Studies

Nine studies [14,15,22,28,29,30,31,32,33] enrolled 222 patients with acute mania/hypomania/mixed symptoms (mean age 37.2 ± 13.0 years, females = 33.7%). Only one [22] of the five RCTs used adjunctive clonidine to lithium treatment while the remaining four studies explored the efficacy of monotherapy with clonidine for mania. Among the non-randomized studies, only one study examined the role of clonidine as monotherapy [33], while the other three studies enrolled participants who received either monotherapy or adjunctive clonidine. A detailed description of the characteristics and inclusion/exclusion criteria of included studies is described in Table 1.

### 3.3. Randomized Controlled Trials: Five Studies (n = 159)

#### 3.3.1. Clonidine vs. Placebo

Of the five RCTs, three studies (*n* = 115) compared clonidine’s efficacy to placebo. Janicak et al. (1989) assessed the efficacy of clonidine monotherapy (0.2–0.8 mg daily dose) in 21 patients (mean age 32 ± 11 years) hospitalized for acute mania [28]. The authors did not find any significant difference between clonidine and placebo in reducing manic symptom severity or a reduction of the BPRS score at 14-day follow-up. There was a significantly higher dropout rate in the clonidine group compared to placebo (75% vs. 44%) [28].

Hardy et al. (1989) compared the efficacy of clonidine (*n* = 12) and placebo (*n* = 12) in 24 patients with acute mania during the study duration of 14 days [29]. The authors allowed the use of droperidol (drinkable or intramuscular 50 mg) for both groups based on the clinical assessment. In this study, the authors utilized an initial dose of clonidine of 0.225 mg/day for the first week, which was increased during the second week to 0.45 mg/day. However, in the case of non-response (only one patient), the dose was allowed to increase up to 0.90 mg/day. There was no difference in the droperidol utilization among the two groups. The authors reported no significant difference in clonidine response rates compared to placebo (64.7% vs. 53.4%, *p* = 0.09) [29].

In a recent study, Ahmadpanah et al. (2022) compared the efficacy of adjunctive clonidine (0.2–0.6 mg/day; *n* = 36) and placebo (*n* = 34) in 70 inpatients (mean age 37.40 ± 11.75 years, 84.29% males) with acute mania. All patients received a standard lithium dose of 900–1200 mg/day. At the end of the 24-day trial, the authors reported a significantly greater reduction in manic symptoms severity from baseline to endpoint for clonidine (YMRS Score = 29.11 ± 5.83 at baseline to 9.81 ± 3.49 at day 24) compared to the placebo group (YMRS Scores = 27.26 ± 5.88 at baseline to 13.62 ± 5.67 at day 24; *p* = 0.002) [22].

#### 3.3.2. Clonidine vs. Control (Lithium or Verapamil; *n* = 44)

Two RCTs [30,31] compared clonidine’s efficacy as an antimanic agent to verapamil and lithium, respectively. In a crossover study, Giannini et al. (1985) compared clonidine monotherapy (17 µg/kg/day) to verapamil (80 mg three times/day) in a 45-day study among 23 white men in a current manic episode [30]. Authors used the BPRS to measure the symptoms. The authors reported that verapamil was superior to clonidine in relieving tension, diminishing grandiosity, attenuating anxiety, excitement, and even reducing the overall BPRS scores. Patients reported notable orthostatic effects with clonidine.

In another crossover study, Giannini et al. (1986) compared clonidine monotherapy (17 µg/kg/day) to lithium carbonate (dose adjusted to maintain serum lithium of 1.2 mEq/L) during a 75-day trial among 24 white men in a current manic episode and who previously had a good response to lithium [31]. Patients were randomized to receive lithium (*n* = 12) or clonidine (*n* = 12) for 30 days, followed by placebo for 15 days during the wash-out phase and subsequently 30 days on either lithium or clonidine. Authors used the BPRS to measure the symptoms. At 30 days, the authors reported a statistically significant improvement in the lithium group compared to the clonidine group (*p* < 0.05). After the crossover phase, interestingly, at the endpoint both groups showed no significant differences in the BPRS ratings [31].

### 3.4. Non-Randomized Studies (n = 63)

Jouvent et al. (1980) conducted one of the first studies to investigate clonidine’s efficacy as an antimanic agent. Authors conducted an open-label study to investigate the antimanic effect of clonidine in eight patients with mania or hypomania. Patients received 0.225 mg to 0.45 mg of clonidine every day [32]. A total of 7/8 patients received diazepam 10 mg as a hypnotic at night; one patient received chlorpromazine 25 mg. Droperidol was added in case of insufficient clinical response to clonidine. Authors reported global and partial improvement in 6/8 patients. Two patients required droperidol, indirectly highlighting the lack of efficacy of clonidine. The authors also noted a significant improvement at day 3 for patients who reached 0.45mg/day dose of clonidine [32]. One patient discontinued clonidine due to hypotension; two patients reported depression symptoms necessitating discontinuation of clonidine after 1 week of treatment.

Giannini et al. (1983) conducted a double-blind study (*n* = 11) in which both treating physicians and patients were told that patients may or may not be receiving the active treatment; however, all the patients received clonidine (17 µg/kg/day) [33]. Patients’ symptoms were assessed at baseline, day 10, and day 25 of clonidine treatment using the Mean Mania Rating Scale score (MMRS). Authors reported that all the patients had a significant reduction in manic symptom severity scores from baseline (MMRS score: 219 ± 26) at both day 10 (MMRS score: 136 ± 13; *p* < 0.001) and day 25 (MMRS score: 80 ± 37; *p* < 0.001) with clonidine monotherapy. The antimanic effect was consistent with no development of tolerance at the end of 25 days of treatment. None of the patients discontinued clonidine due to hypotension.

Hardy et al. (1986) conducted a 2-week, open-label study to investigate the antimanic effect of clonidine (0.45–0.9mg/day) in 24 inpatients (mean age 47 ± 19 years, 41.67% males) admitted for treatment of mania [14]. Patients with hypomania or rapid cycling were excluded. Although clonidine monotherapy was preferred, the authors allowed the use of droperidol (oral 25–50 mg) based on the clinical assessment. Patients were also allowed to continue lithium (*n* = 5) if they had achieved a therapeutic serum level of lithium. At day 14, 11 patients (45.83%) had a good response (>50% improvement); three patients (12.5%) had a partial response (25%–50% improvement); and 10 patients (41.67%) had a poor response (<25% improvement) [14]. Early response on day 5 predicted a final clinical response at the study end. The droperidol prescription was inversely proportional to the clonidine’s efficacy. A good prior response to antipsychotics and a family history of depression in first-degree relatives were predictors of poor response to clonidine [14].

Tudorache and Diacicov (1999) conducted a 1-month, open-label study to investigate the antimanic effect of clonidine (0.45–0.75mg/day) in 20 inpatients (mean age 42 ± 18 years, 40% males) admitted for treatment of mania [15]. Half the patients had mood congruent psychotic symptoms. Patients with hypomania or rapid cycling were excluded. Patients received 0.45 mg clonidine in three divided doses for the first 5 days. If there was a 60% improvement, dose was gradually optimized in 0.15 mg increments to up to 0.75 mg dose by day 10. This dosage was maintained for the next 10 days, and then the dose was gradually reduced to reach a dose of 0.15 mg/day by day 30. Although clonidine monotherapy was recommended, the authors allowed the use of haloperidol based on the clinical assessment. At day 20, 14 patients (70%) had a good response (>50% improvement); four patients (20%) had a partial response (25–50% improvement); and two patients (10%) had a poor response (<25% improvement) based on the Bech–Rafaelsen Mania Scale [34]. This response was maintained throughout the study (day 30). Similar to the study by Hardy et al. [14], authors reported an early response to clonidine as a predictor to the clinical response at the study end. Patients had a good tolerance to clonidine, reporting moderate sedation, and mild hypotension was noted at 0.6–0.7 mg daily doses.

### 3.5. Adverse Events

Side effect data were available in up to eight trials (*n* = 152) [14,15,28,29,30,31,32,33] reporting hypotension and depression as the most common side effects. Of the eight studies, six studies reported hypotension, depression, sedation, edema of lower limbs, insomnia, rash, and aggression. Discontinuation of clonidine was reported in seven patients in two studies: four patients due to hypotension (*n* = 2) and rash (*n* = 2) [28], two patients because of depression, and one following severe hypotension [32].

### 3.6. Dropouts

Ahmadpanah et al. reported that 14 patients (seven each from clonidine and placebo groups) out of 84 randomized patients withdrew from the study before any psychiatric assessment was conducted [22]. Janicak et al. reported a dropout rate of 58% by the end of the first week and 75% by the second week in the clonidine group compared to a 44% dropout rate in the placebo group by the second week. The higher dropout rates in the clonidine group were attributed to intolerable side effects in 33.3% of the patients on clonidine [28]. Jouvent et al. reported three dropouts owing to severe hypotension (*n* = 1) at day 2 and depression (*n* = 2) at day 7 [32].

### 3.7. Methodological Quality and Risk of Bias Assessment

The quality assessment of included studies has been reported in Appendix A. Only one RCT was of good quality (low risk of bias), although there is a concern for attrition bias in the study [22]. The other four RCTs have an unclear risk of bias with no information regarding allocation concealment or sequence generation and are at high risk for attrition bias. Three open-label studies [14,32,33] lacked an adequate follow-up period and did not conduct a blind evaluation for endpoints, while none of the open-label studies provided information on statistical significance of outcomes.

## 4. Discussion

### 4.1. Summary and Interpretation of Findings

We conducted a systematic review of nine studies assessing the efficacy of clonidine for acute mania. We qualitatively summarized results from five RCTs and four non-randomized studies, highlighting inconsistent findings across the studies. Non-randomized studies suggested antimanic efficacy of clonidine monotherapy or as an adjunctive therapy with lithium for acute mania. However, in the RCTs, as a monotherapy, clonidine was not superior to placebo and had lower efficacy than verapamil and lithium. A recently published RCT reported superior antimanic efficacy for clonidine as an adjunctive treatment to lithium compared to placebo. However, the authors underscored a potential selection bias as the study completers had lower YMRS scores at baseline (less severity) and were older than subjects who dropped out after randomization. Hypotension and sedation were the common side effects with clonidine but usually at the higher dosages.

The older RCTs investigating clonidine monotherapy [28,29,30,31] for mania had a higher risk of bias. Two RCTs compared the efficacy of clonidine to verapamil and lithium but lacked a placebo group [30,31]. All open-label trials had a high risk of bias with scores below the global ideal score of 16. Thus, results from these studies must be interpreted with caution. Nevertheless, these studies provided preliminary insights into the tolerability of clonidine. Hypotension and depression were the two most reported side effects that led to discontinuation of clonidine in three patients. Hypotension is a known side effect of clonidine, mediated by its partial alpha-adrenergic agonist action in anterior and posterior hypothalamus and partial alpha-adrenergic antagonist actions in the medulla, which decreases the sympathetic outflow to excitatory cardiovascular neurons. Moderate to severe hypotension could significantly impact clonidine’s tolerability. Higher dose (>0.45 mg) contributed to more prominent hypotension. Depression has been suggested to emerge in 1%–2% of patients on clonidine therapy [35] due to probable reduction in noradrenergic levels.

In contrast to the RCTs, preliminary data from non-randomized studies suggested clonidine’s efficacy for acute mania. However, since these trials used clonidine as both monotherapy and/or adjuvant to mood stabilizers, the evidence for the antimanic effects of clonidine remains unclear. Furthermore, non-randomized studies [14,15,32,33] can often produce greater mean effect estimates than RCTs, which may explain the heterogeneity in the results [36].

We must highlight some factors that may have influenced the overall study outcomes. Clonidine dose ranged between 0.15–0.75 mg/day in most of the positive studies [14,15,22,32], while the negative studies used lower clonidine dosages only up to 0.45 mg except for one RCT approaching a maximum dose of 0.8mg/day [27]. A plausible explanation could be that the antimanic effects of clonidine are obtained only at higher dosages, but further placebo-controlled clinical trials are required to ascertain this hypothesis. Interestingly, patients who had no family history of affective disorders in first-degree relatives and poor response to neuroleptics had a positive response to clonidine, which also has been previously suggested in the catecholamine hypothesis of the affective disorders [37]. In addition, only three studies have excluded rapid cyclers or patients with hypomanic symptoms that have been shown to have a natural course of illness distinct from acute mania.

Our review did not find clonidine monotherapy effective in treating symptoms of acute mania; however, we found limited efficacy when used as an adjuvant to mood stabilizers. There may be a concern for the precipitation of depressive symptoms with clonidine in patients with BD; however, these data were provided by only few of the studies [31,32,33]. In one of the RCTs comparing clonidine to lithium, an increase in depression ratings were observed with clonidine, whereas in the lithium group there was either decline or no change in the depression ratings. In the RCTs, dropout rates were significantly higher in the clonidine group compared to placebo or control group. Thus, raising concerns regarding tolerability at higher dosages. However, the tolerability of clonidine was much better in the non-randomized studies, especially up to a 0.45 mg daily dosage.

A recent umbrella review on drug repurposing as add-on treatments in mania and bipolar depression by Bartoli et al. found out that, in mania, allopurinol and tamoxifen seem to be more effective than placebo, though with low and very low quality of evidence, respectively [38]. Our review shows low grade evidence showing clonidine as an adjuvant agent for mania. Current evidence on drugs repurposed in manic episodes is limited not only for clonidine; however, the evidence base recommendations for actual drug repositioning of clonidine in daily clinical practice to date is limited.

While we did not look specifically into the neurophysiological mechanisms of clonidine in acute mania, previous studies have suggested the role of an increased noradrenergic concentration during a manic state [13]. Clonidine is a partial agonist of the presynaptic alpha-2 central adrenergic receptors and acts by reducing the release of noradrenaline in the synaptic cleft [39]. Furthermore, it down-regulates cellular firing rates via feedback regulation from recurrent axonal and dendritic collaterals [40], which may all contribute hypothetically to attenuating the hyper noradrenergic states in the manic phase of the illness. However, there is a lack of sufficient functional, molecular, and preclinical studies exploring evidence into its potential mechanism of action in alleviating symptoms of mania to warrant its use as monotherapy.

### 4.2. Strengths and Limitations:

Our review has several strengths, including an exhaustive literature search for all major databases and a manual search of all references in the included studies to avoid selection bias.

We should acknowledge several limitations in this systematic review such as the small number of studies, small sample size, and a higher risk of bias in the included studies. The high variability in clonidine dosages (0.15 to 0.9 mg/day), study duration, heterogeneity, and variations in outcome assessments at different time points between the studies limited the possibility of conducting a meta-analysis. All but one study are very old, which probably impacted the study quality, risk of bias, and heterogeneity in the review. Older RCTs were of lower quality and had a higher risk of bias. In addition, we should underscore differences in study duration, which could also have influenced the overall study’s outcome. For example, the mean study duration was of 27.6 days ranging from 7 to 74 days.

## 5. Conclusions

This systematic review suggests low-grade evidence for clonidine as an adjuvant treatment for acute mania with mood stabilizers. Dose dependent hypotension remains the major side effect, although overall it was well tolerated. The current evidence for clonidine’s efficacy as monotherapy in the acute manic phase is insufficient. However, the role of clonidine as an adjuvant treatment warrants further well-designed, larger sample sized RCTs.

## Figures and Tables

**Figure 1 brainsci-13-00547-f001:**
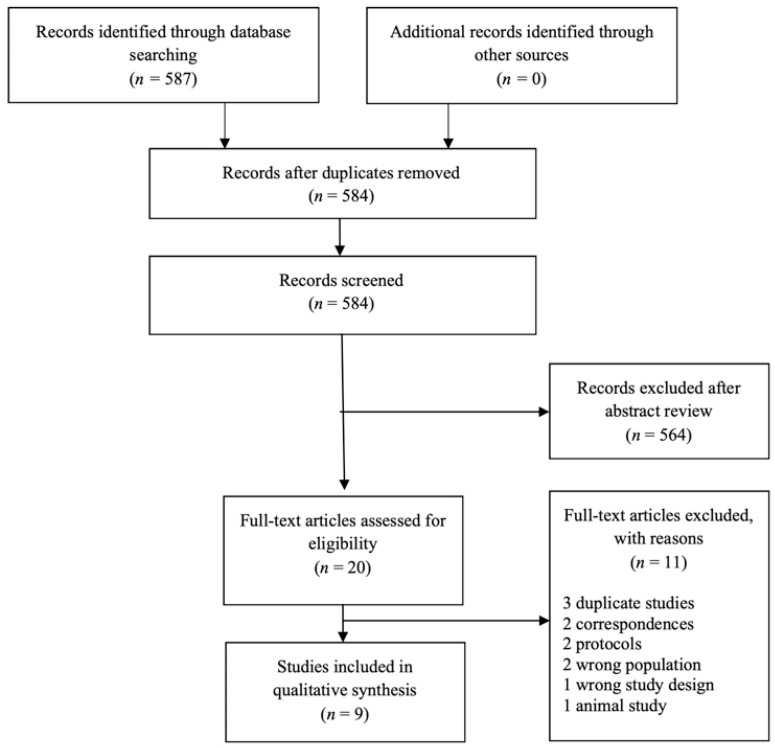
Study selection process.

**Table 1 brainsci-13-00547-t001:** Characteristics of the included trials.

Author, Year, Country	Study Type	Control Group n (Male %)	Clonidine Group n (Male %)	Inclusion Criteria	Exclusion Criteria	Age in Years (Mean ± SD)	Outcome Measures	Study Duration (Days)	Dose of Clonidine	Concomitant Medications Allowed	Conclusions
Jouvent et al., 1980 [32], France	Open Label	NA	8 (62.5)	Adult patients diagnosed with mania or hypomania	Not mentioned	50.5 ± 16.2	Bech MRS; Mean score = 23.5 ± 5.4	7	0.15–0.45 mg/day	Diazepam, Chlorpromazine, Droperidol, Lithium	Clonidine showed antimanic effects. Insomnia, hypotension, and depression were the common side-effects.
Giannini et al., 1983 [33], US	Non-Randomized	NA	11 (54.5)	Research Diagnostic Criteria for mania and hypomania	Concomitant use of psychotropic medication	28 ± 1	Biegel-Murphy MSRS; Mean score = 219 ± 26	25	17 µg/kg/day	NA	Clonidine showed antimanic effects. Relapse observed in patients who discontinued clonidine. Sedation and hypotension were the common side-effects.
Giannini et al., 1985 [30], US	RCT	Verapamil 10 (100)	10 (100)	Adult DSM III for manic episode. Unsatisfactory response to lithium.	Not mentioned	NA	BPRS; Mean score = NA	45	17 µg/kg/day	NA	60% of patients improved with verapamil. Verapamil was superior to clonidine. Hypotension was a prominent side-effect with clonidine.
Giannini et al., 1986 [31], US	RCT	Lithium 12 (100)	12 (100)	Adult DSM III for manic episode. Previous episodes responded to lithium.	Not mentioned	NA	BPRS; Mean score = 1.27 ± 1.02 (lithium); 1.17 ± 1.28 (clonidine)	75	17 µg/kg/day	NA	Lithium showed superior efficacy than clonidine at 30 days. No differences in orthostatic hypotension between groups.
Hardy et al., 1986 [14], France	Open Label	NA	24 (41.7)	Adult DSM III for manic episode. All BD	Rapid cycling; hypomanic	47 ± 19	Bech MRS; Mean score = 26.8 ± 7.1	14	0.45–0.9 mg/day	Lithium, Droperidol	Clonidine showed antimanic effects and an earlier response predicted final outcome. Good tolerability (mild sedation). four patients had worsening of agitation symptoms at higher dose (0.75–0.9 mg).
Janicak et al., 1989 [28], US	RCT	Placebo 9 (NA)	12 (NA)	Adult in-patients diagnosed with BPAD, manic or mixed phase illness according to DSM-III	At least 30% worsening of symptoms on BPRS; intolerable side effects; uncooperative patients	32 ± 18	BPRS, 9-item mania scale; mean BPRS scale = 44.4 ± 10, mean mania scale = 26.6 ± 6	14	0.2–0.8 mg/day	NA	No significant differences between groups. Clonidine had a higher dropout rate
Hardy et al., 1989 [29], France	RCT	Placebo 12 (25)	12 (25)	Adult DSM III for manic or hypomanic episode.	Patients treated with hypertensive agents and associated treatments. Rapid cyclers.	40.8 ± 16.2 (Clonidine group) 30.8 ± 13.6 (Placebo group)	CGS	14	0.25–0.9 mg/day	Droperidol	No significant differences between groups.
Tudorache and Diacicov, 1991 [15], Romania	Open Label	NA	20 (40)	Adult in-patients diagnosed with mania according to DSM-III	Rapid cycling; hypomanic	42 ± 18	Bech MRS; Mean score = 32.4 ± 6.1	30	0.45–0.75 mg/day	Lithium, Haloperidol	Clonidine was efficacious as an antimanic agent. Hypotension was reported in one patient.
Ahmadpanah et al., 2022 [22], Iran	RCT	Placebo 34 (82.3)	36 (86.1)	18–65years of age; Diagnosed with BPAD according to DSM-5; YMRS > 20; At least one hospitalization for BPAD in past 2 years	Known allergies against lithium/alpha-2-adrenergic receptor stimulant drugs. Suicide attempt within the last 8 weeks. Risk of suicide attempt; h/o other psychiatric illness Current use of anticholinergic drugs or TCAs. Females: Pregnant, breastfeeding or planning to become pregnant	37.4 ± 11.75	YMRS; Mean score = 28.21 ± 5.89	24	0.2–0.6 mg/day	Lithium, Sodium Valproate, Antipsychotic	Adjuvant clonidine with lithium significantly improved mania and subjective sleep quality more than placebo, but not cognitive performance.

Abbreviations: BPRS: Brief Psychiatric Rating Scale; CGS: Clinic Global Scale; MRS: Mania rating scale; MSRS: Manic-State Rating Scale; NA: Not applicable; RCT: Randomized Controlled Trial; TCA: Tricyclic antidepressant; YMRS: Young Mania Rating Scale.

## Data Availability

Data are contained within the article or Supplementary Material.

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
