# Peer review of "Efficacy and Safety of Clonidine in the Treatment of Acute Mania in Bipolar Disorder: A Systematic Review"

_brainsci, 2023, doi:10.3390/brainsci13040547_

Round 1

Reviewer 1 Report

Comments and Suggestions for Authors

The article takes up an interesting topic. In addition, it is rather rare that the results and conclusions of the analyses are negative. This shows the scientific and research maturity of the authors. The methodology of the work does not raise objections, the discussion is conducted fairly. The work could have more clearly marked limits. The authors write about it in the discussion and conclusions, but if they had marked it more clearly it would not have closed the discussion on the topic. Finally, it is worth noting that the conclusions directly follow from the analysis.

Author Response

Thank you. We appreciate the helpful comments. We have split the discussion section in two subsections - 4.1 Summary and interpretation of findings and 4.2 Strengths and Limitations.

Reviewer 2 Report

Comments and Suggestions for Authors

This submission reports a methodologically sound and well-written systematic review of randomized and non-randomized studies on efficacy and safety of clonidine (alone or as an adjunctive option) in the treatment of mania. The included studies were synthesized and appraised well, and the Discussion section is quite balanced. 

While this review is surely attractive in view of the growing interest in drug repurposing in psychiatry in recent years, there are some issues and missing points that the Authors should address to make this paper worth of publication:

1. For readers not acquainted with the concept of drug repurposing and its application in neuropsychiatric conditions, I suggest that the Authors provide some background information in their Introduction section. Please refer to Fava’s eminent editorial published in World Psychiatry [https://doi.org/10.1002/wps.20481]. A statement or two (maybe between the first and the second paragraph of the Introduction section, before explaining the rationale for repurposing clonidine) on this matter would be a good fit.

2. The included studies are very old, except for Ahmadpanah et al., 2022, an RCT about clonidine+lithium vs. lithium alone for the treatment of acute mania. Actually, this was also the only one of good quality. Thus, the Authors should:

2a. Report the years when the studies were published in the Results section (3.1 Study Selection subsection).

2b. Among limitations (pages 10-11), the fact that all but one studies are very old (more than 30 years) must be acknowledged and what this implies for their quality and ultimately for the findings of this review should be discussed.

3. To improve readability, the Discussion section may be split in two subsections: 4.1 Summary and interpretation of findings and 4.2 Limitations.

4. According to the findings of this review, how does evidence available on clonidine compare with findings on other drugs repurposed for mania? Indeed, a very recent and relevant umbrella review on drug repurposing as add-on treatments in mania and bipolar depression by Bartoli et al. [2021, https://doi.org/10.1016/j.jpsychires.2021.09.018] found out that, in mania, allopurinol and tamoxifen seem to be more effective than placebo, though with low and very low quality of evidence, respectively. It is really important that the Authors discuss their findings and compare them to those by Bartoli et al., 2021, considering that current evidence on drugs repurposed in manic episodes is limited not only for clonidine, not allowing making reliable recommendations for actual drug repositioning in daily clinical practice to date.

5. The discussion of clonidine-induced hypotension and how it affects tolerability should be expanded.

6. The tables should be reviewed and redesigned. Indeed, Table 2 reports some characteristics of primary studies / populations (inclusion and exclusion criteria) that should be paired with other characteristics reported in Table 1. Moreover, some other interesting characteristics such as manic symptoms severity at baseline should be considered.
7. Table 1, last row: more details should be added, i.e., “significantly more than placebo” and “but not cognitive performance”.

Reviewer 3 Report

Comments and Suggestions for Authors

In this systematic review, authors investigated the efficacy and safety of clonidine’s use as mono- and adjuvant therapy of manic states in patients with bipolar disorder. The present systematic review builds upon the sparse literature data, which includes only 9 studies, 8 of which are over 30 years old, and one recent, published in 2022. Could authors give some explanation about the reason for this time gap between adequate studies dealing with clonidine application in the treatment of BD? What are the guidelines for further studies to meet the necessary criteria for a more rigorous evaluation of clonidine’s use in the treatment of manic states in BD? Could the authors provide more data regarding clonidine’s utility and justifiability in this indication?

I have some minor suggestions for the authors.

1.     The authors should reformulate keywords and include efficacy and safety, and exclude systematic review.

2.     The authors should exclude word often in the sentence and provide the names of the atypical antipsychotics used in the treatment of the bipolar disorder.

The pharmacological management of acute manic episodes often involves using atypical antipsychotics or mood stabilizers like lithium, sodium valproate, and carbamazepine..

3.     The authors should delete the semicolon and use a comma instead in the sentence:

The noradrenergic hyperactivity theory suggests increased activity in the central sympathetic system, abnormal cortical and thalamic dopaminergic/noradrenergic and serotonin production;

4.     The authors should delete the semicolon in the sentence:

Clonidine is a well-established centrally acting anti-hypertensive agent;

5.     The authors should correct the font size:

The risk of bias was assessed for the RCTs using the Cochrane Collaboration’s risk of bias tool

Round 2

Reviewer 2 Report

Comments and Suggestions for Authors

Good revision following my observations. No further comments.